# Comparison of Training Effects of Bounding and Single Leg Jumps for Speed on Sprint and Jump Kinematics in Young Female Football Players

**DOI:** 10.3390/jfmk10040468

**Published:** 2025-12-02

**Authors:** Bjørn Johansen, Jonathon Neville, Roland van den Tillaar

**Affiliations:** 1Department of Sports Sciences and Physical Education, Nord University, 7600 Levanger, Norway; 06091696@student.nord.no; 2Sports Performance Research Institute New Zealand, Auckland University of Technology, Auckland 1010, New Zealand; jneville@aut.ac.nz

**Keywords:** horizontal plyometric training, sprint performance, step kinematics, training adaptation, female youth athletes

## Abstract

**Objectives:** This study compared the training effects of two horizontal plyometric training interventions over six weeks on sprint performance and jump kinematics in young female athletes. **Methods:** Nineteen female football players (age 15.3 ± 0.5 years) were stratified by sprint time into a bounding for speed group (n = 10) or a single leg jumps for speed group (n = 9). All participants completed pre- and post-tests including a 40 m sprint, bounding, and single leg jumps for speed with both legs. Sprint times and velocities over 10 m, 20 m, and maximal speed were recorded, and jump kinematics (horizontal velocity, step length, and step frequency) were analyzed. **Results:** A significant main effect of time was found for sprint performance, indicating that both groups improved overall. The single-leg jump group showed significant within-group improvements across all sprint measures (10 m, 20 m, maximal velocity, and 40 m time) and significant increases in horizontal velocity and step length during the single-leg jump with both legs. The bounding group showed no significant sprint improvements, with only a within-group increase in step frequency during bounding and a trend toward shorter step length (*p* = 0.037, ηp^2^ = 0.40). **Conclusions:** Both training groups improved sprint performance overall, but only the single-leg jump group showed consistent within-group gains in both sprint and jump performance. These findings suggest that single-leg jumps for speed may be a practical and effective option for developing sprint-related qualities in young female football players, although the differences between groups should be interpreted with caution.

## 1. Introduction

Sprinting and rapid acceleration are critical performance determinants in many sports, including football, handball, rugby, and athletics. While maximal velocity is essential in sprint events, the ability to accelerate effectively over the first 10–20 m is particularly important in team sports, where short, high-intensity efforts are performed frequently [1,2]. To improve sprint performance, athletes typically supplement sprint training with various support exercises, including strength and plyometric training [3,4]. Among these, horizontally oriented plyometric exercises such as bounding and single leg horizontal jumps are widely used to enhance force production and coordination for sprinting [5,6]. The rationale for using such exercises lies in the importance of producing force horizontally during sprinting. This is particularly evident in the early acceleration phase, where horizontal force production has been identified as a key determinant for sprint performance [7,8]. However, horizontal force also plays a significant role at maximal velocity, influencing step length, ground contact time, and the ability to maintain speed [9].

Plyometric exercises involving horizontal displacement, such as bounding and single leg horizontal jumps, are believed to offer a high degree of mechanical specificity to sprinting [3,5,10]. These movements challenge the neuromuscular system to produce force rapidly in the direction of running and closely resemble the mechanical demands of sprint strides [7,10]. Studies comparing vertical and horizontal plyometric training suggest that the direction of force application influences the transfer of training adaptations [3,4,6]. Horizontally oriented plyometric training appears to transfer more effectively to sprint performance than vertical jump training, as shown in both single studies [5] and meta-analyses [3]. This is likely due to the greater mechanical and directional specificity of horizontal movements, which more closely replicate the force orientation and movement patterns required during sprint acceleration and maximal velocity running.

Bounding for speed and single leg jumps for speed may therefore be particularly relevant, as they replicate key elements of sprinting such as short ground contact times and forward-oriented propulsion. Bounding also involves alternating limb movements, making it especially similar to the rhythmic stride pattern of sprinting, while single leg jumps challenge strength, balance, and coordination due to repeated unilateral take-off actions on the same limb [10]. Although both exercises involve high-intensity horizontal displacement, they differ markedly in their kinematic profiles, both from each other and from sprinting. During the acceleration phase of sprinting, athletes typically reach velocities of 6–7 m/s, with step lengths between 1 and 2 m, ground contact times of 0.13–0.15 s, flight times of 0.06–0.09 s, and step frequencies of 4.5–5.0 Hz [11,12]. At maximal velocity, elite sprinters often exceed 10 m/s, with step lengths above 2 m and both contact and flight times around 0.10–0.12 s, while frequency remains relatively stable [11,12]. Bounding for speed typically involves velocities of 5–8 m/s, step lengths around 2 m, and step frequencies of 3–4 Hz, with contact and flight times of approximately 0.11–0.20 and 0.15–0.25 s, respectively [12,13]. Single leg jumps for speed show lower velocities (3–5 m/s), longer step lengths (above 2 m), and notably longer contact and flight times (typically 0.20–0.25 s) [13,14] compared with bounding. Although both bounding and single leg jumping are performed over a fixed distance with the goal of maximizing horizontal velocity, they differ not only in kinematics but also in technical demands. Although bounding is biomechanically similar to sprinting in terms of step length, frequency, contact time, and rhythmic alternation between legs, the exercise is technically demanding as it requires coordination of two opposing elements: long steps combined with high frequency. By contrast, single leg jumps are performed at lower velocities and frequency, longer contact times, and unilateral execution, but share the same performance goal as sprinting, achieving maximal horizontal velocity, while placing less emphasis on the simultaneous coordination of high step frequency and long step length, which may reduce the complexity of execution for some athletes. These biomechanical and coordinative differences may also lead to distinct neuromuscular adaptations, particularly in young athletes who are still developing inter-limb coordination and rhythm control [3,5,15].

A previous cross-sectional study by Johansen et al. [13] found that horizontal velocity in single leg jumps and bounding correlated moderately to very strongly with sprint velocity in female athletes, but with clear differences depending on athletic background. Among team sport athletes (football/handball), the correlation between bounding and sprint velocity was moderate to strong (*r* = 0.57–0.68), while single leg jumps showed similar or slightly higher correlations (*r* = 0.50–0.76). In the sprinter group, the association between single leg jumps and sprint velocity was substantially stronger (*r* = 0.83–0.92), whereas bounding showed clearly lower correlations (*r* = 0.39–0.46). These differences suggest that bounding and single leg jumps involve distinct movement patterns and performance components, and that their transfer value may depend on athletic background [13].

Despite widespread use of horizontal jumps in both testing and training, no intervention studies to date have directly compared the longitudinal effects of bounding and single leg jumps for speed. Thus, a clear knowledge gap remains regarding which type of fast horizontal jump exercise, such as bounding for speed or single leg jumps for speed, offers the greatest transfer to sprint performance in team sport athletes. This study specifically targeted young female football players to investigate training effects within a homogeneous and practically relevant population. Previous research has reported sex-specific differences in strength, stiffness, and horizontal force production that may influence responses to plyometric training [3,15]. Therefore, the findings should be interpreted within this population context. Based on this, the aim of this study was to compare the effects of six weeks of training with either bounding or single leg jumps for speed on sprint and horizontal jump kinematics in young female football players. Based on previous findings, we hypothesized that both training interventions would lead to improvements in sprint velocity, jump velocity, step length, and step frequency during the jump tasks. However, we expected single leg jumps to result in greater improvements than bounding, as the exercise imposes fewer technical demands. Unlike bounding, which requires athletes to combine high velocity with long steps, single leg jumps focus more directly on forward propulsion without additional technical constraints, potentially making it easier to perform effectively and more responsive to training in young team sport athletes.

## 2. Materials and Methods

### 2.1. Participants

Based on the review by Rumpf et al. [4], which investigated the effects of plyometric training on sprint performance and reported moderate effect sizes with sample sizes of 7–12 participants, it was concluded that 13 subjects per group would be sufficient. Therefore, a total of 26 moderately experienced female football players were recruited and assigned to two intervention groups (13 in each): a bounding group and a single leg jump group. All players were from the same U16–17 girls’ football team competing in a local youth league in Norway. They trained together three to four times per week and participated in regular league matches during the season but were not part of elite or national programs.

This age group was selected because girls at approximately 15 years of age have typically passed their peak height velocity period, reducing variability related to growth and maturation. Nevertheless, some individual differences in biological maturation may still influence neuromuscular adaptation and responsiveness to plyometric training [3,15]. Group allocation was performed using a staggered procedure based on 40 m sprint times: the fastest participant was placed in the bounding group, the second and third fastest in the single leg jump group, the fourth and fifth in the bounding group, and so on.

All participants were familiar with the exercises, having practiced them for six weeks prior to the tests. All participants and their guardians (for those under 18 years) provided written informed consent after being fully informed about the procedures, potential risks, and benefits of the study. The study was approved by the Norwegian Center for Research Data (NSD) (project number 225529) and conducted in accordance with the latest revision of the Declaration of Helsinki.

### 2.2. Procedure

The study followed a pre–post design, with a training period between the two test sessions. Testing and training were conducted immediately after the end of the competitive season. The same procedures were used for pre- and post-testing. Each participant completed an individual warm-up consisting of approximately 5 min of jogging, followed by an injury-prevention program [16] and two submaximal sprints.

Maximal efforts were then performed in the following order: two 40 m sprints, two bounding trials, and two single leg jumps on each leg, performed consecutively (Figure 1). A 2–3 min rest was given between trials depending on athlete readiness. All tests were performed indoors on a tartan surface, during a period of high training load, as the players had already started their pre-season training for the next year. Participants wore regular running shoes. For the sprint and bounding tests, each trial was initiated from a standing position with a self-selected stance (one foot in front of the other). For the single leg jump tests, participants placed the non-jumping leg in front at the start position.

### 2.3. Training Intervention

Most training sessions were conducted indoors, primarily on artificial turf, although some variation in surface type occurred due to facility access. The training period lasted six weeks, with an average of 10.8 ± 1.3 completed sessions in the bounding group and 11.0 ± 1.7 in the single leg jump group.

The intervention was designed to target horizontal power and sprint-specific adaptations, using bounding or single leg jump exercises performed at maximal or near-maximal effort. Each training session was conducted immediately after the athletes’ regular football warm-up and in conjunction with their scheduled football practices. Sprint efforts were integrated in each session to reinforce sprint-specific neuromuscular adaptations and to maintain movement specificity across the intervention.

Both groups trained twice per week. The bounding group performed repeated bounding strides over 30 m, the single leg jump group performed 15 m of single leg jumps on one leg, immediately followed by 15 m on the other leg. Each training session was concluded with linear sprinting. All repetitions were executed at 95–100% intensity, followed by 90 s of rest (Table 1).

### 2.4. Measurements

To measure sprint and jump performance, a laser device (Noptel Oy, Oulu, Finland) was used to track horizontal movement throughout each test. The laser was aimed at the lower back of each participant and employed a CMP3 distance sensor (sampling rate 2.56 kHz), from which horizontal velocity was calculated. For the sprint runs, additional time measurements were obtained using a Freelap timing system (Freelap SA, Lausanne, Switzerland). Athletes started 1 m behind the first timing gate, so the system recorded time over the final 40 m. During the horizontal jump tests, an infrared optical contact mat (Ergotest Technology AS, Langesund, Norway) was used to register ground contact events. The mat consisted of two 0.87 m long units (“master” and “slave”), each 2.5 cm thick, operating at 1000 Hz. A 5 mm high layer of infrared light detected foot contact by registering interruptions in the beam. The two units were placed 20 m apart, creating a horizontal light field across the running lane (Figure 2). Step length was calculated as the horizontal distance measured by the laser from the start of one contact (detected by the infrared mat) to the start of the next contact. Step frequency was derived from step duration (contact plus flight time), and velocity was obtained directly from the laser. For each jump trial, the 3–4 fastest consecutive steps within the measured distance were selected for analysis to represent steady-state performance. All sensor recordings were synchronized and processed using Musclelab software version 10.200.90.5095 (Ergotest Technology AS, Langesund, Norway).

### 2.5. Statistics

Before the main analyses, the normality of each variable was checked using the Shapiro–Wilk test. A set of two-way mixed-design ANOVAs was then run for each of the main outcome variables: sprint velocity at 10 m, 20 m, and maximal speed, 40 m sprint time, and horizontal velocity, step length, and step frequency during the jump tests. Time (pre vs. post) was used as a within-subjects factor, and training group (bounding vs. single leg jump) as a between-subjects factor. To assess changes within each group, a one-way repeated-measures ANOVA was also performed. These within-group analyses were considered exploratory and used to describe performance changes when no significant group × time interaction was found, and they were not used to claim differences between groups. In addition, main effects of time from the mixed ANOVA were reported to illustrate overall changes across both groups. In case group-by-time interactions reached the significance level (*p* < 0.05), group-specific post hoc tests (Holm–Bonferroni adjusted paired *t*-tests) were calculated. Effect sizes from the ANOVAs are reported as partial eta squared (ηp^2^). Values of ηp^2^ around 0.01, 0.06, and 0.14 were interpreted as small, moderate, and large effects, respectively [17]. All analyses were carried out using JASP software (version 0.18.3.0; University of Amsterdam, Amsterdam, The Netherlands), with significance level set at *p* < 0.05.

## 3. Results

Only 19 participants completed the intervention and were included in the final analysis (10 in the bounding group and 9 in the single leg jump group; Table 2). Seven participants did not complete the intervention: three withdrew without providing a reason, and four withdrew due to injury not directly related to the intervention.

Significant main effects of time were found for sprint performance at all distances (*F* ≥ 4.7, *p* ≤ 0.044, ηp^2^ ≥ 0.21; Figure 3 and Figure 4), except for 20 m (*F* = 3.2, *p* = 0.090, ηp^2^ = 0.16), with higher velocities at the different distances in the post-test. No significant group effects (*F* ≤ 1.76, *p* ≥ 0.21, ηp^2^ ≤ 0.09) or time × group interactions (*F* ≤ 3.58, *p* ≥ 0.075, ηp^2^ ≤ 0.17) were observed for any sprint variable. However, when analyzed per group, the single leg jump group showed within-group improvements in all sprint measures, as all nine athletes improved their performance (*F* ≥ 5.6, *p* ≤ 0.045, ηp^2^ ≥ 0.41), while no significant changes were observed in the bounding group, where three athletes decreased their sprint performance after the intervention (*F* ≤ 2.25, *p* ≥ 0.170, ηp^2^ ≤ 0.20; Figure 3). A main effect of time was significant for all sprint distances, indicating that both groups improved similarly across the intervention period. Since the group × time interaction was not significant, these within-group changes should be interpreted as exploratory observations.

For the bounding task, no significant changes over time (*F* ≤ 0.47, *p* ≥ 0.50, ηp^2^ ≤ 0.03) or between groups (*F* ≤ 3.21, *p* ≥ 0.091, ηp^2^ ≤ 0.16) were found for jump velocity (Figure 4), step length, or step frequency, while significant time × group interactions for step length and step frequency were observed (*F* ≥ 4.65, *p* ≤ 0.046, ηp^2^ ≥ 0.22; Figure 4). Post hoc testing revealed that the bounding group significantly increased step frequency (*p* = 0.037, ηp^2^ = 0.40), with a trend toward reduced step length (*p* = 0.054, ηp^2^ = 0.35), while the single leg jump group did not show significant changes from pre- to post-test (*p* ≥ 0.140, ηp^2^ ≤ 0.25; Figure 5).

For the single leg jumps, significant changes over time were found in jump velocity (*F* ≥ 4.9, *p* ≤ 0.040, ηp^2^ ≥ 0.23), but not for step length or step frequency with left or right foot (*F* ≤ 3.40, *p* ≥ 0.065, ηp^2^ ≤ 0.18). In addition, a significant group effect was observed in step frequency with the right foot (*F* = 5.4, *p* = 0.033, ηp^2^ = 0.24) and a significant time × group interaction for jump velocity with the left leg (*F* = 4.58, *p* = 0.047, ηp^2^ = 0.21). No other significant effects were observed (*F* ≤ 3.89, *p* ≥ 0.065, ηp^2^ ≤ 0.18). Post hoc testing revealed that the single leg jump group increased jump velocity with both feet (Figure 3) and step length with the right leg (*p* ≤ 0.050, ηp^2^ ≥ 0.40), while the bounding group did not show significant changes from pre- to post-test (*p* ≥ 0.144, ηp^2^ ≤ 0.22; Figure 5). As for the sprint results, these within-group changes should be viewed as exploratory, given the absence of consistent interaction effects.

## 4. Discussion

The aim of this study was to compare the training effects of two horizontal plyometric exercises: bounding and single-leg jumps for speed on sprint performance and jump kinematics in young female football players. The results partly supported our hypothesis: while no significant interaction effect, but a relatively large effect size (*p* = 0.075, ηp^2^ = 0.17) was found for 40 m sprint performance, both groups showed general improvement over time. The single-leg jump group showed a statistically significant within-group improvement in sprint performance, whereas the bounding group did not; however, this difference between groups should be interpreted cautiously as exploratory rather than confirmatory. The single-leg jump group also showed gains in velocity and step length during the single-leg jump task, whereas the bounding group showed no significant sprint improvements and only an increase in step frequency during the bounding task. These divergent outcomes suggest that the two exercises elicited different adaptations, with single leg jump having a clearer transfer to sprint performance.

Previous studies have shown that horizontally oriented plyometric exercises, such as bounding, can improve sprint performance due to their mechanical similarity to sprinting [3,5,10]. In the present study, however, this effect was not observed for bounding, despite a structured six-week program with similar sprint-specific elements across groups. This indicates that not all horizontal plyometric exercises transfer equally well to sprint performance in young team sport athletes. The discrepancy may reflect differences in technical demands, coordination requirements, or athletes’ ability to execute the movements effectively.

One reason why only the single leg jump group (all players) improved sprint performance could be that the two exercises place different demands on the body and vary in technical complexity. Single leg jumping involves repeated unilateral take-offs with a clear focus on forward propulsion, and may therefore be easier for young players to perform with stable rhythm and control [10]. Bounding, on the other hand, shares several characteristics with sprinting—alternating steps, short ground contact times, and relatively high velocity—but requires athletes to coordinate long strides with high step frequency simultaneously [12,13]. In our experience, several players found it difficult to perform bounding for speed effectively, even after familiarization and repeated practice during training. This challenge seemed related to the need to coordinate long strides with high step frequency, which likely limited the quality of execution and reduced the overall training effect, as shown by only better 40 m sprint performance for seven of the ten players (Figure 3). In contrast, most players performed the single leg jumps with more stable rhythm and control. Although both exercises are executed unilaterally, the single leg jump requires greater vertical displacement to allow the same foot to reposition for the next contact. This longer flight phase results in higher landing impact and greater force demands during take-off and landing [12,13]. Such mechanical loading, combined with simpler rhythmical execution, may have contributed to the larger improvements in step length and horizontal velocity observed in this group.

The changes observed in jump kinematics also reflected differences in how the two groups adapted to the training. The single leg jump group showed significant improvements in horizontal velocity and step length in the single leg jump, which may indicate better force production or improved technical execution. In contrast, the bounding group increased step frequency and showed a non-significant but large effect size toward shorter step length during the bounding task, suggesting a shift in movement strategy rather than gains in power or speed. The unilateral nature of the single leg jump may also have placed greater emphasis on force production and balance per step, potentially stimulating more specific neuromuscular adaptations. This age group may also display variation in coordination, training background, and neuromuscular maturity, which could influence their response to plyometric training. Therefore, the findings should be generalized with caution to other age groups or performance levels. Although the bounding group showed small numerical improvements in sprint performance, none of the changes were statistically significant, and the overall pattern did not indicate a meaningful training effect after six weeks. Similar findings were reported by Moran et al. [3], who noted that horizontally oriented plyometric training produced variable transfer effects when exercises were technically demanding or performed over short intervention periods. These findings suggest that the single leg jump group improved both the trained task and sprint performance, while the bounding group showed minor changes limited to the bounding task, with no measurable transfer to sprinting ability.

From a practical standpoint, the results of this study suggest that single leg jumps are a more effective and accessible option than bounding when the goal is to improve sprint-related qualities in young team sport athletes. They may also provide a more consistent overload stimulus, whereas bounding requires precise execution at maximal intensity to achieve sufficient overload [3,5]. Based on our findings, single leg jumps for speed appear to be a simple, efficient, and sprint-relevant exercise that coaches can use to help young players develop horizontal force and improve sprint ability over time. Overall, the differences between groups should be interpreted with caution, as they were not supported by a statistically significant interaction effect.

This study has several strengths, but also some limitations that are important to consider when interpreting the results. The training programs were clearly structured, progressed in volume and intensity, and were consistently followed by participants. Group allocation was stratified by pre-intervention 40 m sprint performance, and attendance and supervision were high across all training sessions, strengthening internal validity, although a crossover design would likely have provided even stronger control of inter-individual differences. An important aspect of the study is the choice of exercises. While bounding for speed is already widely used in athletic training [3], single leg jumps for speed have primarily served as test exercises in earlier research [13]. These exercises differ significantly in their execution and the specific qualities they target. To our knowledge, this is the first training study to directly compare their effects on sprint and jump performance.

Despite these strengths, the study has some limitations. As the sample size was based upon sample sizes of previous similar studies, the sample size was not so large to begin with (n = 26) and was further reduced by dropouts not directly related to the intervention, leaving 19 participants for analysis. This could cause type II errors, as probably shown by no significant interaction effects in sprint performance. However, despite the low number of subjects, still some significant interaction effects in jump kinematics were found, indicating that the study had enough subjects to avoid type II errors for these variables. All participants were young female football players, which limits the generalizability of the findings to other populations. At the same time, this homogeneous sample reduced variability related to sex, age, or sporting background, thereby strengthening internal validity. Although the single leg jump group showed clear improvements, the lack of a control group means we cannot rule out the influence of other training or seasonal factors, and it remains uncertain whether the adaptations would be sustained long-term. Other factors, such as sprint training, general physical activity, or seasonal variation, may also have contributed. Finally, six weeks of training might not have been sufficient to elicit full improvements in all participants, particularly for more technical and challenging exercises like bounding.

## 5. Conclusions

After six weeks of training, all players in the single-leg jump group demonstrated clear improvements in sprint velocity and performance in the single-leg jump, whereas the bounding group showed changes mainly in step frequency during the bounding task, with seven of the ten players improving 40 m sprint performance after the intervention. These findings suggest that single-leg jumps for speed may provide a practical and effective method to develop sprint-related qualities in young female football players. However, as the group × time interaction did not reach statistical significance, these results should be interpreted with caution. Future studies should include larger samples and different populations to determine whether the observed differences reflect true training effects, and to explore whether combining bounding and single-leg jumps could produce complementary or additive effects on sprint performance.

## Figures and Tables

**Figure 1 jfmk-10-00468-f001:**
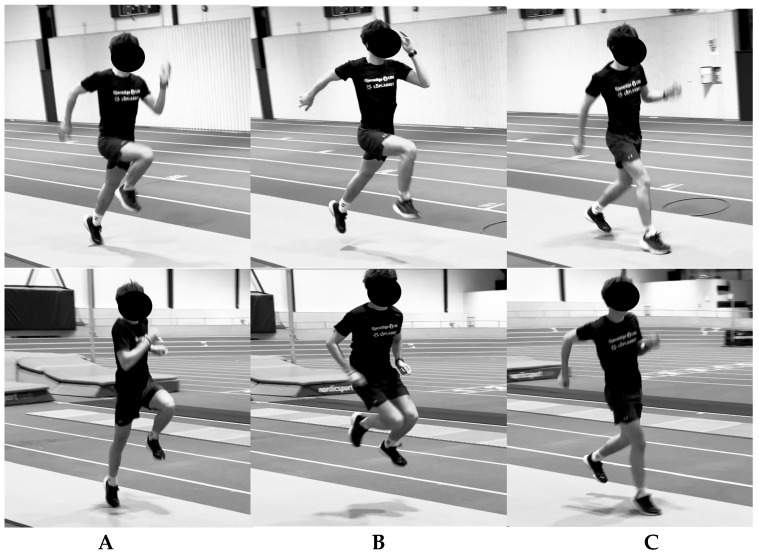
Illustrative movement phases (**A**) take-off, (**B**) flight, and (**C**) landing in bounding (**upper row**) and single leg jump for speed (**lower row**).

**Figure 2 jfmk-10-00468-f002:**
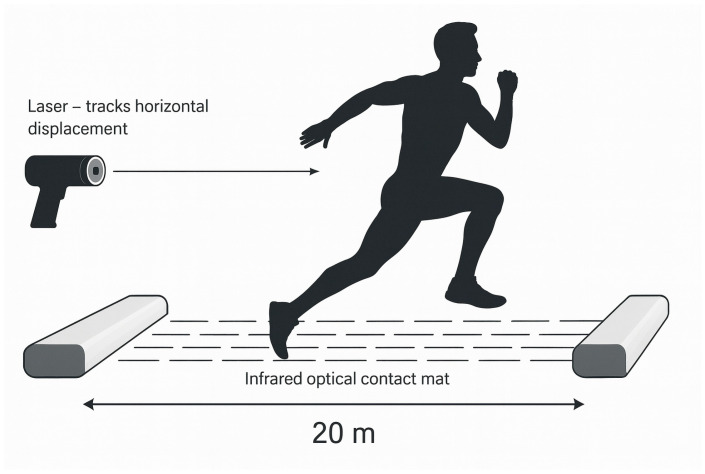
Measurement setup for horizontal jump tests. Two infrared optical contact-mat units were positioned 20 m apart to register foot contacts along the lane, while a laser device placed behind the athlete tracked horizontal displacement and velocity, synchronized in the MuscleLab system.

**Figure 3 jfmk-10-00468-f003:**
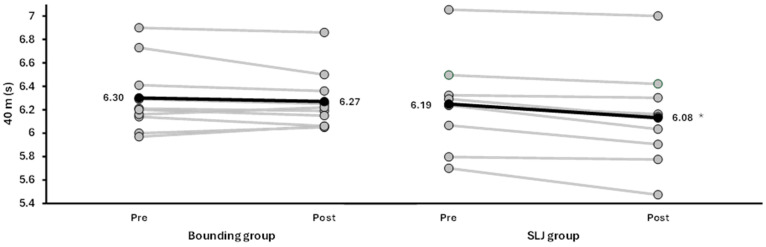
Individual 40 m sprint times (s) before and after the training period for the bounding group (**left**) and the single leg jump group (**right**). Each grey line represents one athlete. Black lines indicate group means, with values shown at each data point. * indicates significant pre–post improvement (*p* < 0.05).

**Figure 4 jfmk-10-00468-f004:**
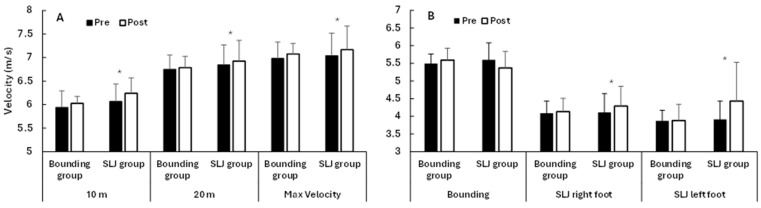
Sprint and jump velocity (mean ± SD) before and after the intervention in the bounding and single leg jump (SLJ) groups. (**A**) Sprint velocity. (**B**) Jump velocity. * indicates significant pre–post improvement (*p* < 0.05).

**Figure 5 jfmk-10-00468-f005:**
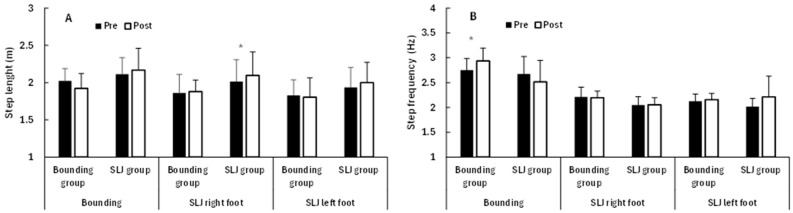
Step length and step frequency (mean ± SD) during horizontal jump tasks in the bounding and single leg jump (SLJ) groups before and after the intervention. (**A**) Step length. (**B**) Step frequency. * indicates significant pre–post improvement within group (*p* < 0.05).

**Table 1 jfmk-10-00468-t001:** Training program for the bounding and single leg jump groups.

Week	Bounding Group	Sprint Addition	Single Leg Jump Group	Sprint Addition
Weeks 1–2	3 × 30 m bounding	3 × 30 m sprint	3 × (15 + 15) m single leg jumps	3 × 30 m sprint
Weeks 3–4	4 × 30 m bounding	3 × 30 m sprint	4 × (15 + 15) m single leg jumps	3 × 30 m sprint
Weeks 5–6	5 × 30 m bounding	3 × 30 m sprint	5 × (15 + 15) m single leg jumps	3 × 30 m sprint

**Table 2 jfmk-10-00468-t002:** Participant characteristics for the bounding and single leg jump groups (mean ± SD).

Variable	Bounding Group (n = 10)	Single Leg Jump Group (n = 9)
Age (years)	15.4 ± 0.5	15.1 ± 0.3
Body height (m)	1.67 ± 0.04	1.73 ± 0.07
Body mass (kg)	60.0 ± 5.8	61.6 ± 6.2

## Data Availability

The data presented in this study are available on request from the corresponding author. The data are not publicly available due to national laws on privacy of the Norwegian government.

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
