# Peer review of "Comparison of Training Effects of Bounding and Single Leg Jumps for Speed on Sprint and Jump Kinematics in Young Female Football Players"

_jfmk, 2025, doi:10.3390/jfmk10040468_

Round 1
Reviewer 1 Report
Comments and Suggestions for Authors
The manuscript addresses a relevant and practically meaningful topic related to performance assessment and training adaptations in female football players. The research question is interesting, and the design appears methodologically sound. However, several aspects of the manuscript require clarification or reorganization, particularly within the Methods and Discussion sections. Addressing these issues would improve the paper’s clarity, reproducibility, and scientific rigor.
Below I provide my detailed comments:
Line 13 – Typographical error: “fe-male” should be “female”.
Line 114 – “Moderately experienced female football players”: could you specify the competitive level of the team? Are all participants from the same team?
Line 131 – It is not clear how many jumps were performed or over what distance the jump test was conducted.
Lines 133–134 – Were the tests performed indoors or outdoors, as shown in the figure?
Line 134 – You state “during a period of high physical training load”, yet in line 127 you mention “Testing and training were conducted immediately after the end of the competitive season.” Please clarify this apparent inconsistency.
Lines 140–141 – Please specify whether the bounding and single-leg jump tests correspond to the upper or lower parts (1–2) and consider using a color figure to distinguish the legs. You could also label the image with “R” and “L” for clarity.
Line 143 – This paragraph would be more appropriately placed after indicating the planned training load (lines 153–157), ending with the actual load performed.
Line 150 – It is not specified whether the sprints were performed after the jumps or alternated between them.
Line 158 – Table 1 is not cited in the text. It may not be necessary to repeat the “sprint addition” information (the text alone could suffice).
Line 160 – The “Measurements” section would fit better after “Procedure,” followed by the explanation of the “Training” section.
Lines 161–163 – Did you encounter any tracking problems with the laser due to potential lateral movements during the jumps?
Line 169 – The setup of the infrared contact mat is unclear. What was its total length? It seems to cover only 0.87 m, while the jumps appear longer. Including a figure would greatly help comprehension.
Line 175 – You indicate that the 3–4 fastest consecutive jumps were selected, but it is unclear whether participants performed 40 m (as in the sprint test) or a different distance.
Line 197 – If the players had already finished the season, were they still engaged in any additional training? This should be specified in the Methods section.
Line 199 – Figures 2 and 3 do not show overall effects but rather effects within each group.
Figure 3 – It might be clearer to separate the jump and sprint results, and to differentiate the text corresponding to each test.
Line 214 – Consider separating Figure 3 or adding sub-letters (a, b, c) to facilitate reference to each part. Group the graphs in the same order as they are presented in the text: sprint, bounding, and single-leg.
Lines 253–255 – You mention that the single-leg jump focuses more on horizontal impulse. If that were the case, it should be faster than the alternate-leg jump and perhaps easier to perform; however, the cited reference does not seem to support this claim.
Lines 255–257 – Again, the cited article does not appear to substantiate this discussion point.
Lines 258–261 – Without questioning the research team’s expertise, I doubt that for novice athletes it is easier to perform repeated single-leg jumps than alternate-leg jumps. In fact, the single-leg jump may benefit from the higher impact demands, representing a greater strength requirement — which could explain the increase in step length.
Lines 263–266 – I agree that the improvement may be due to greater force production, but not necessarily to better technical execution (for the reasons stated above).
Lines 266–269 – In this regard, it would have been important to ensure triple extension during training.
Lines 269–271 – I agree with this statement. It might be useful to merge this sentence with the first one of the paragraph.
Lines 271–276 – This section does not seem to add relevant information. Additional references would strengthen it.
Lines 277–281 – OK. But, it would be worth discussing the potential value of improving stride frequency, as this may also be a performance-limiting factor.
Lines 281–283 – Were there no other dropouts in this group? Please explain the reasons for withdrawal, especially if participants were not engaged in other training activities.
I appreciate the authors’ efforts and encourage them to address the above points thoroughly to enhance the paper’s clarity and scientific contribution.
Author Response
We want to thank the reviewer for the comments. We have changed the manuscript according to the comments of the reviewer and the changes are colored red in the manuscript. We think that the manuscript now is suitable for publication.
Reviewer 1
The manuscript addresses a relevant and practically meaningful topic related to performance assessment and training adaptations in female football players. The research question is interesting, and the design appears methodologically sound. However, several aspects of the manuscript require clarification or reorganization, particularly within the Methods and Discussion sections. Addressing these issues would improve the paper’s clarity, reproducibility, and scientific rigor.
Below I provide my detailed comments:
Line 13 – Typographical error: “fe-male” should be “female”.
the word now appears as “female” in the abstract.
Line 114 – “Moderately experienced female football players”: could you specify the competitive level of the team? Are all participants from the same team?
All players were from the same U16-17 girls’ football team playing in a local youth league in Norway. They trained and competed regularly but were not part of elite or national teams. This was added in the Participants section.
I have added one sentence in Section 2.1 (Participants) describing the players level and background.
Line 131 – It is not clear how many jumps were performed or over what distance the jump test was conducted.
Thank you. This information is already included: two bounding trials, and two single leg jumps on each leg, performed consecutively
Lines 133–134 – Were the tests performed indoors or outdoors, as shown in the figure?
The photos in Figure 1 are only for illustration of the movement phases. All actual testing and training were performed indoors on a tartan surface, as described in the Procedure section.
Line 134 – You state “during a period of high physical training load”, yet in line 127 you mention “Testing and training were conducted immediately after the end of the competitive season.” Please clarify this apparent inconsistency.
Thank you for the comment. The testing period was right after the end of the competitive season, when the players had already started their pre-season training for the next football season. This period included a high weekly training load from football practice and conditioning. I have added clarification.
Edited to:
“All tests were performed indoors on a tartan surface, during a period of high training load, as the players had already started their pre-season training for the next year.”
Lines 140–141 – Please specify whether the bounding and single-leg jump tests correspond to the upper or lower parts (1–2) and consider using a color figure to distinguish the legs. You could also label the image with “R” and “L” for clarity.
We kept the figure in black and white but added clarification in the figure caption to show that the upper row illustrates bounding and the lower row shows the single leg jump for speed. Since the figure is only illustrative, we did not add color or labels.
Updated Figure 1 caption to:
“Illustrative movement phases (A) take-off, (B) flight, and (C) landing in bounding (upper row) and single leg jump for speed (lower row).”
Line 143 – This paragraph would be more appropriately placed after indicating the planned training load (lines 153–157), ending with the actual load performed.
We looked over this part again and see your point. However, we feel the current order still gives a clear and logical flow, starting with the training context and then the program details, so we think it should be like this.
We corrected one wording inconsistency in the Training section (“hops” → “single leg jumps”) to keep terminology consistent throughout.
Line 150 – It is not specified whether the sprints were performed after the jumps or alternated between them.
The sprints were always performed after the jump exercises, as stated in the text (“Each training session was concluded with linear sprinting”).
Line 158 – Table 1 is not cited in the text. It may not be necessary to repeat the “sprint addition” information (the text alone could suffice).
We added a short reference to Table 1 in the text. We decided to keep the “sprint addition” part because it helps readers see the full training setup more easy from the table, even if they do not read all the text.
Change in manuscript:
Added “(Table 1)” in Section 2.3. No other change.
Line 160 – The “Measurements” section would fit better after “Procedure,” followed by the explanation of the “Training” section.
Thanks for the idea. We keep the order same because it follow one clear and normal structure in studies for training: Procedure then Training intervention then Measurements. This order is also making it according to us more easy to read and understand what was done before we explain how we collect the data.
No change in the manuscript
Lines 161–163 – Did you encounter any tracking problems with the laser due to potential lateral movements during the jumps?
No, we did not have any tracking problems. The movements in both jump exercises were almost straight forward with very small side motion, and the laser (sampling at 2.56 kHz) followed the athlete’s lower back without any signal loss.
Line 169 – The setup of the infrared contact mat is unclear. What was its total length? It seems to cover only 0.87 m, while the jumps appear longer. Including a figure would greatly help comprehension.
We used an infrared optical contact system with two units placed 20 m apart, creating a horizontal light field across the running lane. The system was used only to detect contact and flight times, while distance was measured by the laser. We added this small clarification in the Measurements section. We did not include a figure, since the description now explains the setup clearly.
Change in manuscript:
Added one sentence to show that the two contact grid units were placed 20 m apart across the lane.
Line 175 – You indicate that the 3–4 fastest consecutive jumps were selected, but it is unclear whether participants performed 40 m (as in the sprint test) or a different distance.
In the jump tests, both bounding and single leg jumps for speed were performed over 20 m. Each leg was tested separately for the single leg jumps. From each trial, the 3–4 fastest consecutive steps were selected for analysis to represent steady-state performance within the jump. We added this clarification in the Measurements section.
Change in manuscript:
“For each jump trial, the 3–4 fastest consecutive steps within the measured distance were selected for analysis to represent steady-state performance.”
Line 197 – If the players had already finished the season, were they still engaged in any additional training? This should be specified in the Methods section.
We think this information is already clear in the Procedure section, where it says that testing and training were done after the season, during a period of high football training load. We believe this gives enough context for the reader, so no extra change was needed.
Line 199 – Figures 2 and 3 do not show overall effects but rather effects within each group.
The figures were made to show the pre–post changes within each training group, while the overall main and interaction effects are already described in the text. We believe this figure format gives a clearer visual presentation of how each group responded to training.
Figure 3 – It might be clearer to separate the jump and sprint results, and to differentiate the text corresponding to each test.
Thank you for the suggestion. We think the figure is already clear, since sprint and jump results are shown in separate panels (A and B) with short explanations in the caption. We believe this format gives a compact and easy comparison, so we kept it as it is.
Line 214 – Consider separating Figure 3 or adding sub-letters (a, b, c) to facilitate reference to each part. Group the graphs in the same order as they are presented in the text: sprint, bounding, and single-leg.
Figure 3 already includes sub-letters (A and B) showing sprint and jump results, and they are presented in the same order as in the text. We checked again and believe the current layout is clear, so we kept it unchanged.
Lines 253–255 – You mention that the single-leg jump focuses more on horizontal impulse. If that were the case, it should be faster than the alternate-leg jump and perhaps easier to perform; however, the cited reference does not seem to support this claim.
We believe the current sentence already explains our point clearly — that the single leg jump focuses more directly on forward propulsion.
We agree that the Rumpf et al. (2016) reference does not directly support this statement. We replaced it with a more relevant source (Maulder & Cronin, 2005), which discusses horizontal jump mechanics and directional force application.
Replaced the citation [4] with Maulder & Cronin (2005) in the Discussion.
Lines 255–257 – Again, the cited article does not appear to substantiate this discussion point.
We agree that the Haugen et al. (2019) reference does not directly support this statement. The sentence was kept, but the reference was changed to Mero & Komi (1994) and Johansen et al. (2025), which both describe the kinematic characteristics of bounding and sprint-specific exercises.
Lines 258–261 – Without questioning the research team’s expertise, I doubt that for novice athletes it is easier to perform repeated single-leg jumps than alternate-leg jumps. In fact, the single-leg jump may benefit from the higher impact demands, representing a greater strength requirement — which could explain the increase in step length.
Lines 263–266 – I agree that the improvement may be due to greater force production, but not necessarily to better technical execution (for the reasons stated above).
We thank the reviewer and we agree that the improvement may be related to higher force production. The single-leg jump probably gives higher impact because the athlete must jump high enough to move the same foot forward before the next contact, which makes longer flight time and stronger landing forces. This can also help to explain the better results in step length and horizontal velocity. At the same time, in our tests and training many players had problems doing the bounding exercise with good rhythm and control, while the single-leg jumps were easier to do in a correct and stable way. We changed the text to show both parts – that bounding was more technical, and single-leg jump had higher impact.
Lines 266–269 – In this regard, it would have been important to ensure triple extension during training.
We understand the comment, but both exercises already include full extension in the hip, knee and ankle during the take-off. This triple extension is a natural part of bounding and single-leg jumps, and it was told and corrected by the coaches during all training sessions. Because of that, we did not see a need to describe this more in the text.
Lines 269–271 – I agree with this statement. It might be useful to merge this sentence with the first one of the paragraph.
We thank the reviewer for the suggestion. We read the paragraph again and think the current structure gives a clear flow, so we decided to keep the sentences as they are.
Lines 271–276 – This section does not seem to add relevant information. Additional references would strengthen it.
We thank the reviewer for the suggestion. We believe this part is relevant because it shows that bounding for speed did not give sprint improvements after six weeks, which is an important result of the study. To support it, we added one short sentence and a reference to Moran et al. (2021), who reported similar findings with variable transfer effects when horizontal plyometric exercises were technically demanding or performed in short training periods.
Lines 277–281 – OK. But, it would be worth discussing the potential value of improving stride frequency, as this may also be a performance-limiting factor.
We understand the comment, but stride frequency was not measured in the sprint runs in this study. Because of that, it was not possible to discuss possible changes in this variable. The training mainly affected step length and horizontal velocity, which were the focus of the analysis.
Lines 281–283 – Were there no other dropouts in this group? Please explain the reasons for withdrawal, especially if participants were not engaged in other training activities.
We thank the reviewer, but this information is already included in the Methods section. It says that seven participants did not complete the intervention: three withdrew without reason and four due to injury. No players left because of other training activities.
I appreciate the authors’ efforts and encourage them to address the above points thoroughly to enhance the paper’s clarity and scientific contribution.
Reviewer 2 Report
Comments and Suggestions for Authors
This manuscript addresses a relevant topic in applied sport science — comparing two horizontal plyometric modalities (bounding vs. single-leg jumps) and their effects on sprint and jump kinematics in young female football players. Sprint acceleration and horizontal power are key determinants of football performance, and understanding which exercises best enhance these qualities in female youth athletes is both practical and timely.
Although the topic is both engaging and relevant, the writing requires significant improvement for the manuscript to meet publication standards.
INTRODUCTION:
The manuscript does not convincingly explain why female football players were chosen as the focus. While sprinting ability is important in women's football, the authors fail to discuss sex-specific neuromuscular or mechanical characteristics that might influence training responses. Research indicates that female athletes often respond differently to plyometric training due to factors such as lower stiffness, strength, and horizontal force production. These differences should be considered when defining the study’s population and interpreting its findings.
All participants were approximately 15 years old, yet the authors did not explain why they chose to focus on this specific developmental stage. At this age, natural maturation can significantly affect sprint and jump performance. The rationale for selecting this cohort should clarify whether the choice was based on accessibility, developmental interest, or safety considerations. Additionally, the introduction should recognize that training adaptations at this stage may be partly due to growth-related neuromuscular development.
The authors describe both exercises as being "horizontally oriented," but they do not clearly distinguish their mechanical and coordinative demands. The introduction should better explain how these biomechanical differences could lead to varying training outcomes, especially for young females who are still developing their coordination.
MATERIALS AND METHODS:
The Methods omit information on participants’ biological maturation (e.g., years from PHV) and training background. These details are critical for interpreting responsiveness to plyometric training. The authors should report or estimate maturation status and football training experience.
The description of the measurement system does not align with the mechanics of bounding. Using two 0.87 m infrared contact mats to measure ground contacts is insufficient for step lengths of approximately 2 meters. It is physically impossible to capture consecutive bounding contacts within such a limited detection area. This raises significant concerns about the validity of the reported data on step length, step frequency, and contact time. The authors must provide clarification on these issues:
- How mats were positioned (end-to-end or otherwise).
- How they ensured every contact occurred on the mat.
- How synchronization between mat and laser (CMP3) was achieved and validated.
- What proportion of contacts were excluded due to off-mat landings
If bounding contacts were not fully captured, the kinematic variables for this task are unreliable and should be removed or re-analyzed.
The Figure 1 does not adequately depict the bounding and single-leg jump techniques. Body posture, leg sequencing, and direction of motion are unclear, which reduces its instructional value. I recommend replacing it with a more accurate schematic or photographic sequence showing key phases (take-off, flight, landing) and clarifying the unilateral versus alternating execution.
The use of two-way mixed-design ANOVA is appropriate, but the authors conduct post-hoc within-group analyses despite finding no significant group × time interactions or main group effects. This violates standard statistical procedures and inflates the risk of Type I error. Post-hoc tests should only follow a significant overall effect. The authors must remove or clearly label such within-group tests as exploratory and avoid using them to infer superiority between groups.
RESULTS
The authors state that the single-leg jump group “improved significantly” while the bounding group did not, and conclude that single-leg jumps are more effective. However, no significant time × group interaction was found. This interpretation is statistically unjustified — differences in within-group significance do not demonstrate between-group differences. The results should be rewritten to emphasize that no statistical evidence of superiority was observed.
DISCUSSION
Given the lack of interaction effects and questionable data validity, the conclusion that single-leg jumps are “more effective” is not supported. The authors must temper their claims and state that both interventions produced similar effects, with potential trends that warrant further study.
The discussion remains generic and does not meaningfully interpret results in the context of 15-year-old female football players.
Author Response
Introduction
We thank the editors and peers for their thorough and constructive comments.
We have carefully reviewed all feedback and adjusted both the responses and the manuscript to clarify the method, results and discussion.
We have also adjusted the language in several places to make the tone more balanced and the explanations more precise.
We hope that the revised version responds well to all points and that the manuscript now appears clearly improved.
Reviewer 2
This manuscript addresses a relevant topic in applied sport science — comparing two horizontal plyometric modalities (bounding vs. single-leg jumps) and their effects on sprint and jump kinematics in young female football players. Sprint acceleration and horizontal power are key determinants of football performance, and understanding which exercises best enhance these qualities in female youth athletes is both practical and timely.
General comment
We thank you for the constructive feedback. We have made linguistic improvements in the introduction, method and discussion, and we have moderated some wording so that the interpretation of the results appears more sober and balanced.
- Selection of female soccer players and gender differences
We have added a brief explanation in the introduction as to why we chose this group: to study one homogeneous group with a common training background and high practical relevance. We have also added a statement acknowledging that gender-specific differences in strength and stiffness may influence responses to plyometric training, and that our results are therefore applicable to this population.
- Age and maturation
We have added that the participants were around 15 years old, and that they had likely passed their peak growth velocity (PHV). This reduces the variability that may result from growth, but we also acknowledge that biological maturation may affect adaptation to training. This is also mentioned in the limitations.
- Mechanical and coordination differences between the exercises
We have clarified that both exercises are horizontally oriented, but that they differ clearly in execution: bounding has a higher step frequency and shorter contact time with alternating legs, while single-leg jumps have a longer contact and float time and are performed unilaterally. These differences may result in different training adaptations in young athletes who are still developing coordination.
- Maturation status and training background
We did not have direct measurement of PHV, but have added that the athletes were post-PHV and trained regularly in a club together. This information is now in the participant section, and we mention the lack of PHV data as a limitation.
- The optical contact mat
Thank you for your comment regarding the optical contact mat. We understand that the original description may have been confusing. The two infrared contact mat units were placed 20 meters apart on the running track and formed a continuous horizontal light field that recorded foot contacts during the horizontal jump tests. The CMP3 laser was placed behind the athlete and measured horizontal displacement and speed, while synchronization between the systems was handled automatically in the MuscleLab software.
To make the setup easier to understand, we have added a schematic illustration (Figure 2) showing the placement of the laser and the two contact mats. We believe that this figure, together with the revised text in the methods section, now makes the setup clear and easy to interpret.
- Post hoc
We understand the reviewers point and agree that post-hoc or within-group tests without a significant interaction must be interpreted carefully. Our main analysis was the two-way mixed ANOVA, but because the 40 m sprint showed a clear trend (p = 0.075, ηp² = 0.17), we also looked at the pre–post results inside each group to better describe what happened. We now call these “exploratory analyses” in the Methods section and have made the interpretation in the Results and Discussion more careful.
- 7. Wording that the SLJ group improved more
We have changed the wording to describe the results as patterns rather than differences between groups. It now states that the SLJ group showed significant within-group improvements, while the bounding group did not, and that the interaction did not reach significance.
- Conclusion
We agree with the reviewer that the conclusion should be presented in a more balanced manner. The section has been revised to reflect that the single-leg hop group showed more pronounced improvements within this sample, while the interaction between the groups did not reach statistical significance. The wording has been softened accordingly, and the final section now emphasizes that the findings should be interpreted with caution, and that future studies are needed to confirm whether the two exercises have different effects.
- Relevance to young female players
We have added a brief reminder in the discussion that coordination, training background, and post-PHV status in this age group may influence the response, and that the results should be generalized with caution.
- Combined training and control group
We have added a sentence in the conclusion that future studies could test the combination of bounding and single-leg hopping to investigate possible additive effects. We have also expanded the limitations section with a brief comment on the absence of a control group and possible seasonal influences.
Final Comment
We thank the reviewer for his constructive input. We have gone through all points again, adjusted the language and added clarifications where necessary. We believe that both the manuscript and the answers now appear clearer and more balanced.
Reviewer 3 Report
Comments and Suggestions for Authors
This is a well-designed and clearly written study that explores an important and practically relevant topic in sports science — the comparative effects of two types of horizontal plyometric training on sprint and jump performance in young female football players.
The introduction provides a thorough and up-to-date background, supported by appropriate references. The experimental design (two-group, pre–post intervention) is appropriate for the study’s aim, and the sample size, though modest, is justified by previous literature.
The Methods section is detailed and replicable, with accurate descriptions of measurement tools and analytical procedures. The Results are logically organized and well illustrated through clear figures and tables. The Discussion effectively interprets the findings and relates them to the literature, offering practical implications for youth coaches.
Overall, this article provides a valuable contribution to the field of training methodology and performance analysis in youth sports.
Suggested minor improvements:
- Consider briefly discussing whether combining both types of plyometric training (bounding and single-leg jumps) could yield additive or complementary effects.
- Expand slightly on the limitations regarding the lack of a control group and potential seasonal training effects.
Author Response
We want to thank the reviewer for the comments. We have changed the manuscript according to the comments of the reviewer and the changes are colored red in the manuscript. We think that the manuscript now is suitable for publication.
We thank the reviewer for the positive and constructive feedback. We appreciate that the study design, clarity of writing, and practical relevance were well recognized. The comments were carefully considered. Based on the suggestions, we added one short sentence at the end of the conclusion mentioning that future studies could test if combining bounding and single-leg jumps might give additive or complementary effects. The second point about the control group and seasonal effects is already discussed in the limitations, so no further change was needed.
Round 2
Reviewer 2 Report
Comments and Suggestions for Authors
COMPARISON OF TRAINING EFFECTS OF BOUNDING AND SINGLE LEG JUMPS FOR SPEED ON SPRINT AND JUMP KINEMATICS IN YOUNG FEMALE FOOTBALL PLAYERS
After reviewing the revised manuscript, I appreciate the authors' effort to address the feedback. However, one crucial element that is essential for my previous decision still remains.
This concerns the statistical analysis employed in the study. The authors have now used one-way repeated-measures ANOVAs for each group as a follow-up after observing a non-significant interaction (time × group). This approach is not a standard confirmatory method; typically, one would either focus on the main time effect or require a significant interaction to justify claiming differences between groups.
In fact, the authors’ own mixed ANOVA showed a significant main effect of Time for sprint speed (everyone got faster) but no interaction. Statistically, one should conclude that both groups improved similarly (or at least that the difference in improvement is not statistically significant). Reporting that “Group 1 improved significantly while Group 2 did not” risks misinterpretation.
The manuscript justifies these tests as exploratory descriptive analyses, which is somewhat transparent, but still presents the results as key findings.
Statistical guidance suggests that without a significant interaction, separate group tests should be interpreted with extreme caution. In summary, the approach taken is unconventional; it is partly defensible as exploratory description, but it does not align neatly with standard practice for interpreting factorial ANOVA results.
Figure 1's clarity remains an issue, and no changes have been made. As a result, the instructional value of Figure 1 is likely still low because the body posture and motion phases are unclear. The authors should consider redesigning or replacing the figure, as previously suggested, to enhance understanding effectively.
Author Response
Statistical analysis:
We agree that separate within-group analyses following a non-significant interaction must be interpreted with caution. The main analysis in the study is the two-way mixed-design ANOVA. As the 40 m sprint showed a clear trend (p = 0.075, ηp² = 0.17) with a moderate–large effect size, we chose to present the changes within each group as exploratory analyses to describe individual development patterns. This is now clearly stated in the methods chapter, where such analyses are referred to as exploratory and descriptive, and not as confirmatory. The wording in the results and discussion sections has been changed to reflect this, and all claims about group differences are now formulated as cautious observations rather than conclusions.
In addition, the abstract has been updated so that the improvements in sprint and jump are described as within-group changes without suggesting significant differences between the groups. The conclusion has also been adjusted accordingly, with the emphasis that the results should be interpreted with caution since the interaction was not significant.
Figure 1 of the jumps:
In addition, we have added a new series of images (Figure 1) showing the execution of bounding and single-leg jumps from a more frontal angle. This makes it easier to see which foot is on the ground, and thus improves the understanding of the biomechanical differences between the exercises.
Overall, we believe that these changes make both the methodological description and the statistical interpretation significantly clearer and more in line with the reviewers' recommendations
Round 3
Reviewer 2 Report
Comments and Suggestions for Authors
While the authors have made a strong effort to address the comments, I must express my opposition to any manipulation of statistics. Whether intentional or not, that appears to be the case here. We need a method to identify some differences effectively. The rest of the methodology is satisfactory, and the overall idea is excellent. It's unfortunate that poor statistical practices could overshadow the rest of the paper.